# The FlagT4G Vaccine Confers a Strong and Regulated Immunity and Early Virological Protection against Classical Swine Fever

**DOI:** 10.3390/v14091954

**Published:** 2022-09-02

**Authors:** José Alejandro Bohórquez, Miaomiao Wang, Ivan Díaz, Mònica Alberch, Marta Pérez-Simó, Rosa Rosell, Douglas P. Gladue, Manuel V. Borca, Llilianne Ganges

**Affiliations:** 1WOAH Reference Laboratory for Classical Swine Fever, IRTA-CReSA, 08193 Bellaterra, Spain; 2Unitat Mixta d’Investigació IRTA-UAB en Sanitat Animal, Centre de Recerca en Sanitat Animal (CReSA), Campus de la Universitat Autònoma de Barcelona (UAB), 08193 Barcelona, Spain; 3IRTA, Programa de Sanitat Animal, Centre de Recerca en Sanitat Animal (CReSA), Campus de la Universitat Autònoma de Barcelona (UAB), 08193 Barcelona, Spain; 4Veterinary Diagnostic Laboratory, College of Veterinary Medicine, University of Illinois at Urbana-Champaign, Urbana, IL 61820, USA; 5Departament d’Acció Climàtica, Alimentació i Agenda Rural, Generalitat de Catalunya, 08007 Barcelona, Spain; 6Plum Island Animal Disease Center, Agricultural Research Service, United States Department of Agriculture Greenport, Greenport, NY 11944, USA

**Keywords:** vaccine efficacy, CSFV, innate immunity, FlagT4G, marker vaccine, virological protection, antibody response

## Abstract

Control of classical swine fever virus (CSFV) in endemic countries relies on vaccination, mostly using vaccines that do not allow for differentiation of vaccinated from infected animals (DIVA). FlagT4G vaccine is a novel candidate that confers robust immunity and shows DIVA capabilities. The present study assessed the immune response elicited by FlagT4G and its capacity to protect pigs for a short time after vaccination. Five days after a single dose of FlagT4G vaccine, animals were challenged with a highly virulent CSFV strain. A strong, but regulated, interferon-α response was found after vaccination. Vaccinated animals showed clinical and virological protection against the challenge, in the absence of antibody response at 5 days post-vaccination. Upon challenge, a rapid rise in the titers of CSFV neutralizing antibodies and an increase in the IFN-γ producing cells were noticed in all vaccinated-challenged pigs. Meanwhile, unvaccinated pigs showed severe clinical signs and high viral replication, being euthanized before the end of the trial. These animals were unable to generate neutralizing antibodies and IFN-γ responses after the CSFV challenge. The results from the present study assert the fast and efficient protection by FlagT4G, a highly promising tool for CSFV control worldwide.

## 1. Introduction

Classical swine fever (CSF) is one of the most relevant diseases in animal health, posing a serious threat to the porcine industry worldwide, as well as to food security [1,2]. The causing agent, CSF virus (CSFV), is a member of the Pestivirus genus in the Flaviviridae family [3]. Due to its severe impact, CSFV is a mandatory notification to the World Organization for Animal Health (WOAH, formerly OIE) [4].

Vaccination and stamping out policies against CSFV have been carried out for decades and have led to successful eradication in western Europe, North America, and Oceania [5,6,7]. Despite these extensive efforts, the disease remains endemic in Asia and large parts of Central and South America, including recent outbreaks in Colombia, Russia, Korea, and Japan. Notably, the re-emergence of CSFV in Japan in 2018, where the disease had been eradicated over two decades ago, shows the continuous threat that neighboring endemic countries pose to CSF-free territories [8,9,10,11]. One of the reasons contributing to the continued circulation of the virus in vaccinated populations is the ineffective application of vaccines in the field. To understand the continued circulation of CSFV in endemic areas under vaccination, viral evolution studies have been conducted. Previous studies showed that prolonged suboptimal vaccination programs may have caused changes in the pathogenicity and antigenicity of the new emerging strains that could potentially escape vaccination [12,13,14,15,16]. Therefore, there is a need for novel CSFV vaccine candidates that afford fast and robust immune responses against currently circulating viral strains.

In addition to an efficient immune response, an appropriate CSFV vaccine candidate, according to the standards required by the WOAH, should comply with the differentiation of infected from vaccinated animals (DIVA concept) [17]. Thus far, the development of these vaccines has been centered around the E2 glycoprotein, with subunit and live-attenuated virus vaccine candidates having been generated [18,19]. The diagnostic differentiation for these vaccines has been centered on using a specific ELISA test for the detection of antibodies against the E^rns^ glycoprotein [20,21]. However, an increasing number of reports have pointed out issues with the specificity of this diagnostic test, as it cross-reacts with other pestiviruses, posing a serious concern about its applicability in field conditions [22,23,24].

A promising live-attenuated DIVA vaccine candidate, named FlagT4G, has been developed based on the CSFV Brescia strain [25]. This vaccine candidate has been genetically modified to carry a mutation in an epitope within the most immunogenic viral protein, the E2 glycoprotein. Moreover, the FlagT4G virus also has an insertion of a FLAG peptide sequence. Recently, a DIVA serological test has been developed to fulfill the DIVA capabilities of the FlagT4G vaccine candidate [26].

FlagT4G has proven to induce clinical protection against CSFV challenges as early as 3 days post-vaccination [27]. Taking this into account, the aim of the present study was to assess the virological protection against challenges with a highly virulent CSFV strain, conferred by the FlagT4G vaccine candidate after 5 days of vaccination in domestic pigs, and to elucidate the humoral and cellular immune mechanisms behind the protection afforded by the vaccine candidate.

## 2. Materials and Methods

### 2.1. Cells and Viruses

The porcine kidney cell line PK-15 (ATCC-CCL-33) was grown in Eagle’s Minimum Essential Medium supplemented with 5% fetal bovine serum (FBS). This cell line was used for viral production, as well as for titration and neutralization assays. The CSFV FlagT4G vaccine virus and the highly pathogenic Margarita strain (genotype 1.4), were used in the in vivo and viral neutralization assays. Viral titers were determined by end-point dilution, calculated following standard statistical methods [28], with immune peroxidase monolayer assay (IPMA) being employed for viral replication monitoring [29].

### 2.2. Experimental Design

Ten pigs, at five weeks of age, were introduced into the biosafety level 3 (BSL-3) facilities at IRTA-CReSA, Spain. The animals were purchased from a pestivirus-free farm and had been proven to be free of antibodies against CSFV, prior to entering the facilities. Animals were randomly divided into two groups of five pigs each (Group A and B) and housed in two different pens within the same room. After one week of acclimation period, Group A animals (pigs 1–5) were vaccinated with the CSFV FlagT4G vaccine at a dose based on a previously published study (10^5^ TCID_50_) [27], via intramuscular injection in the right neck, whereas group B pigs (numbered 6–10) served as unvaccinated controls. At 5 days post-vaccination (dpv), all the animals were challenged with 10^5^ TCID_50_ of the highly pathogenic CSFV Margarita strain. After viral challenge, daily clinical monitoring, including rectal temperature, was carried out by a trained veterinarian in a blinded manner. In accordance with previous studies [7], a clinical score value from 0 to 6 was assigned daily to each animal: 0, no signs; 1, mild fever; 2, mild to moderate clinical signs; 3, moderate clinical signs; 4, moderate to severe clinical signs; 5, severe clinical signs; and 6, death. Serum and nasal and rectal swab samples were collected on the day of vaccination, day of viral challenge, and at 4, 7, and 14 days post-challenge (dpc). Whole blood sample was also taken from all the pigs on these dates for the collection of peripheral blood mononuclear cells (PBMCs). At the end of the trial (14 dpc), all the animals were euthanized, and tonsil, spleen, and mesenteric lymph node samples were obtained. Euthanasia was performed by a pentobarbital overdose of 60–100 mg/kg of weight, administered via the jugular vein, in accordance with European Directive. Pigs were also euthanized before the end of the trial for animal welfare reasons if they reached a clinical score of 5. The experiment was carried out according to existing Spanish and European regulations and was approved by the Ethical Committee of the Generalitat de Catalonia, Spain under the animal experimentation project number 10908.

### 2.3. Antibody Detection by ELISA and Virus Neutralization Assay

Commercial ELISA tests were used to detect antibodies against the E2 (IDEXX Laboratories, Liebefeld, Switzerland) and E^rns^ (INDICAL bioscience GmbH, Leipzig, Germany) glycoproteins, in all serum samples collected. In accordance with the manufacturer’s instructions, for the CSFV E2-specific antibodies, samples were considered positive when they showed a blocking percentage ≥40%. In the case of the CSFV Erns-specific test, sample/positive (S/P) values ≥0.5 were determined to be positive.

Additionally, neutralizing antibody titers against CSFV Margarita and FlagT4G strains were assessed in all serum samples using a neutralization peroxidase linked assay (NPLA) [30]. The neutralization titres were expressed as the reciprocal dilution of serum that neutralized 100 TCID_50_ in the 50% of the culture replicates.

### 2.4. Evaluation of the Cytokine Response by ELISA and Luminex Assays

Serum samples were tested by a previously described in-house ELISA for the detection of interferon (IFN)-α concentration [31,32] on the day of vaccination and at 5 dpv (day of challenge), 4 dpc, and 7 dpc. Serial dilutions of recombinant IFN-α protein (PBL Biomedical Laboratories, Piscataway, NJ, USA) were employed as a standard in the ELISA. The optical density of the standard was used to perform a regression curve, used to quantify the concentration of IFN-α in sera, with results being expressed as units/mL.

In addition, serum samples were analyzed at day of vaccination, 5 dpv (day of challenge), 4 dpc, and at the time of euthanasia (14 dpc), using the cytokine and chemokine 9-Plex Porcine ProcartaPlex™ Panel 1 (nineplex, Affymetrix, eBioscience, Santa Clara, CA, USA). This assay was used for the evaluation of nine different cytokines: IFN-α, IFN-γ, interleukin (IL)-1β, IL-4, IL-6, IL-8, IL-10, IL-12, and tumoral necrosis factor (TNF)-α. Results were quantified on a Luminex^®^ 200™ and a regression curve for serial standard dilutions of each cytokine was obtained, in accordance with the manufacturer’s instructions. Cytokine concentrations were reported as picograms/milliliter (pg/mL), based on their mean fluorescence intensity.

### 2.5. Detection of CSFV RNA

Viral RNA was extracted from all the serum, tissue macerate, and swab samples from all pigs, using the MagAttract 96 cador Pathogen Kit (Qiagen), from an initial sample volume of 200 µL. Tissue samples had been previously ground in 900 µL of Eagle’s Minimum Essential Medium, with the supernatant being used for extraction after centrifugation at 13,000 RPM for 10 min. After extraction, the RNA was stored at −80 °C, until analysis by the specific pan-CSFV RT-qPCR test [33], as well as the specific Margarita strain RT-qPCR [34]. Reactions were performed in a final volume of 20 µL, using the AgPath-ID™ One-Step RT-PCR Reagents (applied biosystems, Waltham, MA, USA): RT-PCR buffer (1x), RT-PCR Enzyme mix (1x), forward primer (0.6 µM), reverse primer (0.6 µM), probe (0.1 µM) and RNA template (4 µL). The temperature profile was 10 min at 48 °C, 10 min at 95 °C, followed by 40 cycles of 2 s at 97 °C and 30 s at 61 °C. Samples were considered positive when cycle threshold (Ct) values were below 40. Samples were also characterized as having high (Ct below 22), moderate (Ct between 23 and 28) and low (Ct between 29 and 40) viral RNA load, as previously described [31,35].

### 2.6. PBMC Collection

PBMC separation was carried out by density-gradient centrifugation using Histopaque 1.077 (Sigma-Aldrich, St. Louis, MO, USA), followed by osmotic shock to lyse the remaining erythrocytes. After two washes with PBS, PBMC was diluted in FBS, supplemented with 10% DMSO, and frozen until use.

### 2.7. Assessment of Cellular Immune Response by IFN-γ ELISPOT

The cellular response, in terms of IFN-γ production by PBMC, was evaluated using an ELISPOT assay, as previously described [35,36]. Briefly, pre-wetted filter plates (MultiScreen-HA filter plate MAHAS4510, Merck, Rahway, NJ, USA) were coated overnight at 4 °C with capture antibody P2G10 directed against porcine IFN-γ (BD Biosciences Pharmingen, Franklin Lakes, NJ, USA) diluted in 0.05 M carbonate-bicarbonate buffer pH 9.6. After washing and blocking, thawed PBMC were seeded at a rate of 5 × 10^5^ cells/well. The cells were stimulated with either the FlagT4G vaccine or the CSFV Margarita strain, at a multiplicity of infection (MOI) of 0.1. Mock-stimulation with culture media was carried out as negative control and used as a baseline for spot quantification of each sample, while stimulation with phytohaemagglutinin (PHA; 10 µg/mL) was used as a positive control. Each animal and assay were carried out in duplicate. After 48 h incubation, detection antibody P2C11 (BD Biosciences Pharmingen) diluted in PBS was dispensed into the wells and the reaction was revealed by incubation of plates with streptavidin–peroxidase and addition of insoluble TMB (Sigma-Aldrich). Average counts of spots in unstimulated wells were subtracted from average counts in antigen-stimulated wells and results were expressed as responding cells/10^6^ PBMC.

### 2.8. Phenotypical Profile Evaluation by Flow Cytometry

The phenotypical profile of immunologic cell subsets in the PBMCs from all the animals was evaluated on the day of vaccination, day of viral challenge (5 dpv), and at the time of euthanasia, using flow cytometry. Hybridoma supernatant, kindly provided by Dr. J. Dominguez (INIA, Madrid, Spain), was used for staining of the CD172a (BA1C11, IgG1), with an anti-Mouse IgG1 antibody labeled with Alexa Fluor 647 (thermofisher scientific, produced in goat) used as a secondary antibody. Conjugated antibodies were also used for the detection of porcine CD4 (Alexa Fluor^®^ 647 Mouse Anti-Pig CD4a 74-12-4, IgG2b, BD Pharmingen), and CD8-α (FITC Mouse Anti-Pig CD8a 76-2-11, IgG2a, BD Pharmingen).

Briefly, 10^6^ cells/well were seeded in a V-shaped bottom cell culture plate (Costar). After removing the media, 50 μL of the corresponding antibody was added to each well, followed by incubation at 4 °C × 30 min. Subsequently, the cells were washed and, in the case of the CD172a staining, the secondary antibody was added. In the CD4 and CD8 staining wells, FACS buffer (PBS + 2% FSB) was added. After incubation at 4 °C × 30 min, covered from light, the cells were washed and resuspended in FACS buffer. Propidium iodide was used as a viability control, added at a final concentration of 1 μg/mL before passing the cells through the cytometer. Only live-cell events (20,000/sample) were considered for subsequent analysis. Assays were carried out in duplicate. The FACSDiva software, version 6.1.2 (BD Biosciences, San Jose, CA, USA), was employed for data visualization and analysis, and the results were expressed as the percentage of positive cells obtained for each staining. The gating strategy used for flow cytometry analysis is provided in Appendix A.

### 2.9. Statistical Analysis

Statistical analysis was performed using the StatsDirect v2.7.7 software (StatsDirect Ltd., Birkenhead, UK). The Kruskal–Wallis non-parametric test was used for comparisons of means between groups; the Friedman test was used for comparisons of means within the same experimental group.

## 3. Results

### 3.1. FlagT4G Affords Early Clinical Protection against CSFV Challenge in Pigs

Group A pigs, vaccinated with FlagT4G, showed a fever peak after FlagT4G vaccination and prior to the CSFV challenge. In all cases, the fever did not last for longer than 3 days and was resolved by 1 dpc (Figure 1A). After the CSFV Margarita challenge, only one of the vaccinated animals showed mild fever (40.3 °C), for one day, at 7 dpc (pig 3), while pig number 4 had mild diarrhea between 1 and 6 dpc. The remaining group A pigs did not show any clinical signs throughout the trial. On 7 dpc, pig 2 suffered a cardiac arrest during sampling and could not be resuscitated, being removed from the study.

Conversely, four out of the five pigs in group B started to show mild fever by 2 dpc. The following day, all the pigs in this group developed a fever (up to 41.7 °C) and showed apathy (Figure 1). By 4 dpc, one animal (pig 6) was showing moderate apathy and mild tremors, accompanied by dyspnea. The clinical picture of these animals continued to worsen and at 6 dpc, one of the pigs showed prostration behavior and had to be euthanized. Finally, at 7 dpc, all the remaining pigs in group B had reached a clinical score of 5 and were euthanized for animal welfare reasons.

### 3.2. FlagT4G Induced Fast Protection after Five Days of Vaccination in the Absence of Antibody Response

None of the FlagT4G-vaccinated animals (group A) were positive for antibodies against CSFV prior to the Margarita challenge, by any of the antibody detection tests employed. Following the viral challenge, at 4 dpc, pigs 2 and 4 were positive in the commercial CSFV-E2 assay and at 7 dpc, four out of five vaccinated animals were clearly positive, reaching the highest levels of detection at 14 dpc (Figure 2). In the E^rns^ ELISA, pigs 1 and 2 showed antibody response at 4 dpc, which continued to increase until the end of the trial for pig 1, whereas pig 2 was negative in the subsequent sampling (Figure 2). Pig number 3 was also positive by this assay at 14 dpc.

After the viral challenge, the NPLA results showed that all the group A pigs, except for pig 1, had neutralizing antibody titers against FlagT4G by 4 dpc, ranging from 1:10 to 1:20, whereas only pig 4 showed antibodies against the Margarita strain (titer 1:10) on this date. By 7 dpc, all the pigs in this group had antibody titers between 1:30 and 1:80 against FlagT4G, which increased towards the end of the trial. In the Margarita strain NPLA, pigs 3 and 4 were positive at 7 dpc, with titers of 1:15 and 1:20, respectively. At 14 dpc, all the surviving pigs had a neutralizing antibody response against the Margarita strain, with titers ranging from 1:10 to 1:80 (Table 1).

By contrast, none of the animals in the infection control group showed detectable antibody response throughout the trial, by any of the assays performed.

### 3.3. Cytokine Response after FlagT4G Vaccination and CSFV Challenge

All the group A pigs showed increased IFN-α levels in serum sample following vaccination with FlagT4G, with concentration values ranging from 278.3–314.8 units/mL at 5 dpv, as detected by ELISA test. The cytokine concentration was greatly decreased after this time and only low IFN-α levels (below 20 units/mL) were detected in two animals during the rest of the trial. In the infection control group, IFN-α concentration in serum peaked at 4 dpc, with values near or above 300 units/mL in all the pigs from this group. By the time of euthanasia, between 6 and 7 dpc, the values had decreased in all the group B pigs, but still ranged from 148.9 to 305.2 units/mL.

In the Luminex assay, a similar pattern to the ELISA test was detected, with significantly increased IFN-α levels in sera after vaccination in group A pigs (between 130.8–203.6 pg/mL at 5 dpv, *p* < 0.05), followed by a steep decrease (Figure 3B). In the case of group B animals, the CSFV challenge also resulted in a significant increase in IFN-α concentration (ranging from 85.7 to 271.8 pg/mL on 4 dpc, *p* < 0.05), compared to pre-challenge levels. This was followed by a decrease at the time of euthanasia in four of the five group B pigs, except for pig number 8, which showed an increase in IFN-α levels, by Luminex, at 7 dpc (Figure 3B and Appendix A).

In the group B pigs, significantly higher IL-12 levels were also detected after CSFV challenge (*p* < 0.05), ranging from 362.9 to 1187.9 pg/mL at 4 dpc, compared to the baseline concentrations before the challenge (between 65.8 and 403.4 pg/mL), (Figure 3B).

Differences between the experimental groups, in terms of cytokine response, were also detected throughout the trial. Vaccinated pigs showed higher concentrations of IL-1β than the infection control group at the day of challenge and at 4 dpc (*p* < 0.05). The TNF-α and IL-4 concentration in sera were also increased in the group A animals compared to group B, on the day of the CSFV challenge. Values ranged from 2.1 to 36.5 pg/mL for IL-4 and between 6.9 to 169.4 for TNF-α in the vaccinated animals, while these cytokines were not detected in any serum samples from group B at this time point (Figure 3B). Conversely, group A animals had significantly lower levels of IL-6 than group B pigs after the CSFV challenge (4 dpc, *p* < 0.05). No statistical differences were detected either in IL-8 or IL-10 at any time point during the trial. All the samples were negative for IFN-γ levels in sera at all the evaluated time points (Data not shown).

### 3.4. The FlagT4G Vaccine Induces Rapid Protection against Systemic Viral Replication

Following vaccination with FlagT4G, low viral RNA load was detected in the serum samples from all vaccinated animals (group A) at 5 dpv, using the pan-CSFV RT-qPCR, with Ct values between 29 and 35. In addition, two pigs were positive on nasal swabs and four more were positive on rectal swabs, all of them with Ct values corresponding with low viral RNA load (Figure 4A).

After challenge with the highly virulent Margarita strain, four of the group A pigs were positive by the pan-CSFV RT-qPCR in serum at 4 dpc, as were two of them in rectal swab sample, with Ct values corresponding with low RNA load. After this time point, only sera from pig 2 at 7 dpc and rectal swabs from pig 3 at 14 dpc were positive by this assay (Ct values >35). Both samples were negative by the Margarita-specific RT-qPCR, as were all sera and swab samples from vaccinated pigs. Therefore, only one sample, pig 3 at 7 dpc was positive for the Margarita strain RT-qPCR test (low levels), according to its Ct value (38.42) (Figure 4B).

By contrast, all but one of the samples from group B animals were positive at 4 dpc, by the pan-CSFV molecular assay, with Ct values ranging from 28.02 to 36.24. In the Margarita-specific assay, all the serum and rectal swab samples were positive at this time point and showed higher RNA load, with Cts between 26.11 and 34.97. The viral RNA load, as detected by both RT-qPCRs, increased at 7 dpc, with all the pigs showing moderate CSFV RNA load in all the samples analyzed and high Margarita RNA load in serum and rectal swabs (Figure 4).

In the analyzed tissues, Ct values from all the samples of the vaccinated pigs corresponded with low to moderate viral RNA load, as detected by pan-CSFV RT-qPCR. In the Margarita-specific assay, three tonsil samples were negative and the remaining two showed moderate to low RNA load. Meanwhile, high CSFV RNA load was detected in all the tissue samples from group B animals, according to their respective Ct value, in both molecular assays employed (Figure 4).

### 3.5. FlagT4G Vaccination Improves Cellular Immune Response in Swine

In the ELISPOT assay, no IFN-γ response was detected in the PBMCs from either of the experimental groups on the day of the CSFV challenge (5 dpv). At 4 and 7 dpc, all the group A pigs (FlagT4G-vaccinated) showed IFN-γ producing cells, against both FlagT4G and the challenge strain (Figure 5), while PBMCs from group B animals remained unresponsive to either of the viral stimuli. All the experimental animals showed an adequate response to the mitogenic agent, used as a positive control (PHA, data not shown).

The phenotypical profile of PBMCs, evaluated by flow cytometry showed a higher proportion of CD8^+^ cells in the vaccinated animals at the time of the CSFV challenge (5 dpv), while the unvaccinated pigs had slightly higher CD4^+^ cells and CD172^+^ cell subsets. After the CSFV challenge, both experimental groups showed a decrease in the percentage of myelomonocytic cells (CD172^+^), with this cell population being significantly lower and nearly undetectable in the unvaccinated pigs at the time of euthanasia (*p* < 0.05). The CD8^+^ cells population was also altered at this time point, showing a significant increase in the unvaccinated pigs, compared to the pre-challenge sample (*p* < 0.05) (Figure 6).

## 4. Discussion

The eradication of CSFV continues to be an elusive task for developing countries, where suboptimal vaccination practices have led to the emergence of attenuated variants that go unnoticed in the field and may have vaccination escape capabilities [1,12,37]. Nevertheless, vaccination remains the best alternative for disease control, considering the economic and ethical costs of stamping out policies [38,39]. Therefore, the generation of a vaccine candidate that provides fast and robust immunity, while complying with the current standards of DIVA diagnosis, is a highly important tool to achieve control of the disease.

The present study shows the capacity of FlagT4G, a novel vaccine candidate with potential DIVA capabilities [25,26], to induce efficient protection against highly virulent CSFV challenges as fast as 5 dpv. The vaccinated pigs were protected against clinical disease, only showing very mild clinical signs that were resolved within the first days after the viral challenge. Moreover, FlagT4G largely protected the pigs against viral replication after challenge with the highly virulent Margarita strain, as shown by the negative results in the Margarita-specific RT-qPCR assay (Figure 4B). In this regard, the protection afforded by FlagT4G against CSFV viral replication was comparable to that elicited by the C-strain or Thiverval vaccines at 5 dpv [7,40,41].

Interestingly, the protection conferred in the vaccinated pigs at 5 dpv was not dependent on antibody response, since all the animals were negative by ELISA and NPLA tests at the time of challenge (Table 1 and Figure 2). Previous studies have shown a correlation between the induction of T cell responses and the protection against CSFV in the absence of antibody response [40,42,43]. Despite the absence of detectable antibodies before the CSFV challenge, neutralizing antibody titers were clearly detected, as early as 4 dpc in the FlagT4G vaccinated pigs. This suggests the presence of low levels of memory B cells in vaccinated animals that could be rapidly boosted, upon viral challenge [42]. The CSFV neutralizing antibodies were increased at later points after the challenge in FlagT4G vaccinated–challenged pigs, while no antibodies were found in non-vaccinated–challenged animals during the trial (Table 1). The level of CSFV neutralizing antibodies induced upon challenge were in agreement with the clinical and virological protection conferred. It should be noted that the early protection afforded by FlagT4G, was robust in terms of virological protection considering the short time after vaccination. However, two of the tonsil samples from the vaccinated group of pigs were positive for the detection of Margarita strain RNA. In addition, the fact that the vaccine strain RNA was found in sera and body secretions of vaccinated-challenge pigs indicates that the vaccine strain may be replicating in the host. However, the Ct values detected in swab samples from these animals corresponded with viral RNA levels below the threshold for the viral isolation [44]. In addition to the abundant clinical signs and the development of the acute form of CSF in the unvaccinated–challenged animals, CSFV RNA could be detected by the specific RT-qPCR to detect the Margarita strain challenge virus from all the different samples analyzed (Figure 4B). On the other hand, the absence of detectable CSFV antibody response after 5 dpv, renders the DIVA ELISA test unable to differentiate the animals in the present study, considering its properties as a negative marker ELISA [26].

IFN-γ producing cells play an important role in the immune response against CSFV, being used as an indicator of efficient induction of immunity in vaccine candidates, even at short times post-vaccination [45,46]. However, in this study, no IFN-γ producing cells were detected after five days post FlagT4G vaccination. Interestingly, IFN-γ producing cells were detected as early as 4 dpc, in direct contrast to what is seen in the unvaccinated animals (Figure 5). This detection is probably improved by the boost effect of the viral challenge on the immune response, previously stimulated by the vaccine. These results suggest that other mechanisms of the T cell response, probably of innate characteristics, are mediating the protection conferred by the vaccine and further studies will be carried out to clarify them.

From the panel of cytokines analyzed in the present study, the only cytokine that showed significantly higher expression in both experimental groups was the IFN-α (Figure 3B). It is noteworthy that an increase in the IFN-α response was detected after five days of FlagT4G vaccination. This cytokine is a well-known immunomodulator, capable of inducing an antiviral state in the host cells, aiding to control infection and leading to the establishment of adaptive immunity [31,47]. This appeared to be the case in the vaccinated pigs, all of which went on to develop a robust antibody and adaptive cellular response, that likely played a role in the protection against the constant viral challenge posed by the high viral secretion of the unvaccinated pigs. This is in line with findings previously reported for the early protection afforded by the C-strain [41] or the Thiverval strain against CSFV infection [7].

On the other hand, an exaggerated IFN-α response has also been shown to be responsible for causing the clinical signs associated with CSF [44,47,48], as was likely the case in the unvaccinated animals from the present study after Margarita infection. In these pigs, the exacerbated innate immune response unravels a cascade of events, such as tissue damage, which lead to the lethal outcome before the onset of adaptive immunity [1,44,49]. It should be noted that the IFN-α levels induced by FlagT4G vaccination of animals in group A were similar to those detected in pigs of group B after the challenge with a highly virulent CSFV strain (Figure 3A). However, in the vaccinated pigs, the IFN-α response was rapidly regulated, whereas non-vaccinated-challenge pigs continued to show high levels of this cytokine at the time of euthanasia. These results show the importance of immune response regulation for the outcome of CSFV infection. No significant changes in lymphocytic or myelomonocytic subsets were detected neither after CSFV vaccination nor challenge in the vaccinated pigs. The lack of variation in these cell subsets from these animals indicates that immune homeostasis is being maintained and the immune response is being properly regulated, even after the highly virulent challenge. This is also evidenced in the cytokine response, which showed similar concentrations within the vaccinated group before and after the CSFV challenge. Conversely, the unvaccinated pigs showed an increased CD8^+^ cell subset after CSFV infection (Figure 6), even though these cells probably were not functional, considering the lack of IFN-γ producing cells in these animals. It is more likely that this increase in CD8^+^ cells leads to a decrease in the CD4/CD8 ratio in unvaccinated animals. This ratio is used as an indicator of altered immune states in other viral infection models and has been previously found in CSFV persistently infected animals [50,51,52,53].

Notably, despite showing antibodies against the E2 glycoprotein and neutralizing antibody titers, the majority of the FlagT4G vaccinated pigs were negative for antibodies against E^rns^ throughout the trial (Figure 2B). This assay has been proposed as a DIVA diagnostic technique for subunit vaccine candidates based on the CSFV E2 glycoprotein [21,38], in which vaccinated animals would be negative for anti-E^rns^ antibodies, while CSFV infected pigs would be positive. Recently, this test has been shown to cross-react with antibodies generated by other pestiviruses able to infect pigs, compromising its specificity [24,54]. The results from the present study show that, in addition to specificity, the detection of anti-E^rns^ antibodies may also have issues in terms of sensitivity and its implementation as a DIVA test may not be fully reliable.

Taken together, the results from the present study assert the capacity for fast and efficient protection afforded by the FlagT4G vaccine, in terms of clinical and virological protection against highly virulent CSFV challenges. Likewise, these results open the possibility of using this vaccine in an emergency vaccination strategy. Further studies will provide insight into safety issues, as well as other aspects (protection in pregnant sows and against vaccine escape variants, among others) that must be addressed. Nevertheless, FlagT4G poses a highly promising tool for CSFV control worldwide, given its efficient immunogenic capabilities, as well as the joined development of its corresponding DIVA assay.

## Figures and Tables

**Figure 1 viruses-14-01954-f001:**
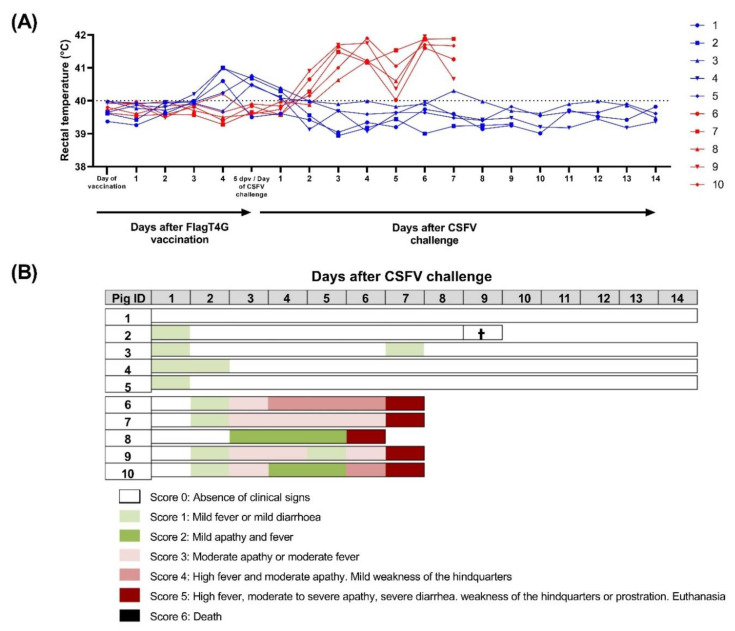
Clinical signs and rectal temperature in pigs during the trial. Individual rectal temperature (**A**) was recorded in FlagT4G-vaccinated (blue lines and symbols) and unvaccinated (red lines and symbols) pigs throughout the trial. Values above 40 °C (dotted line) were considered as fever. (**B**) A numeric clinical score value (represented in colorimetric scale) was assigned for each animal after CSFV challenge. Cross symbol indicates an animal that died during monitoring, despite not showing clinical signs.

**Figure 2 viruses-14-01954-f002:**
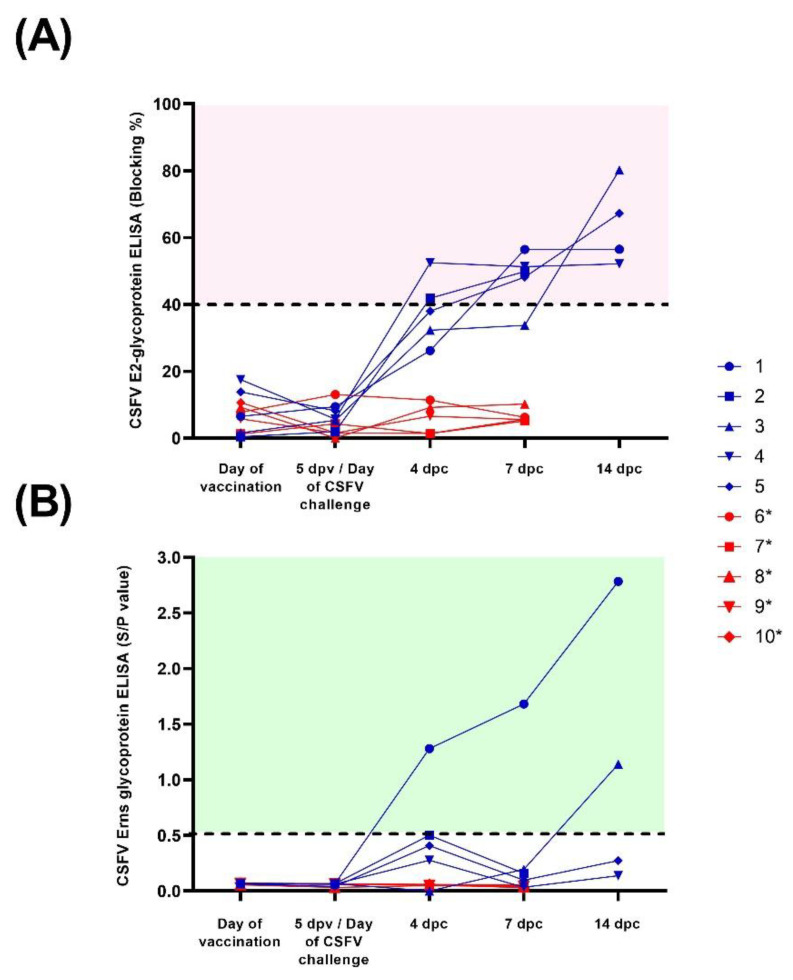
Antibody response after CSFV vaccination and challenge. Sera samples from FlagT4G-vaccinated (blue lines and symbols) and unvaccinated (red lines and symbols) animals were collected and evaluated for antibodies against the E2 (**A**) and E^rns^ (**B**) glycoproteins. For the E2 ELISA test, results are expressed as blocking percentage, and values above 40% (red-shaded area) were considered positive. In the case of anti-E^rns^ antibodies, results were expressed as the sample/positive value with samples above 0.5 (green shaded area) determined to be positive. * Euthanized for animal welfare reasons.

**Figure 3 viruses-14-01954-f003:**
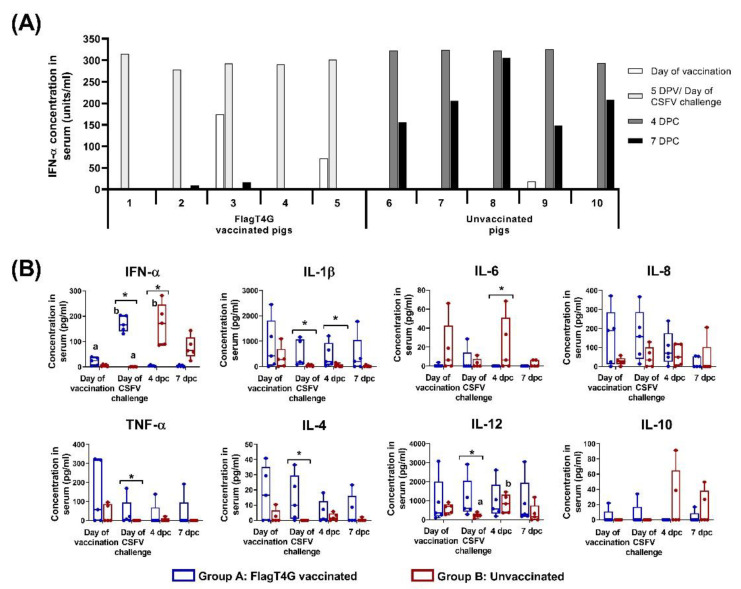
Cytokine response after CSFV vaccination and challenge. (**A**) IFN-α concentration in sera samples was evaluated after vaccination and CSFV challenge, using an in-house ELISA test. Individual results for each sampling are represented as the mean value (in units/mL) of duplicates performed for each sample. (**B**) Sera samples from vaccinated (blue boxes) and unvaccinated (red boxes) animals were also evaluated by Luminex assay for the detection of nine different cytokines. Mean values for each group at every sampling time are represented and cytokine concentrations are shown as pg/mL. Asterisk indicates statistically significant differences between groups on that particular sampling (Kruskal–Wallis non-parametric test; *p* < 0.05). Differences between a given sampling and the previous one within the same group (Friedman test) are represented by the letters on top of the bar: different letters show statistical difference (*p* < 0.05). Not represented: IFN-γ concentration in sera (not detected in any sample).

**Figure 4 viruses-14-01954-f004:**
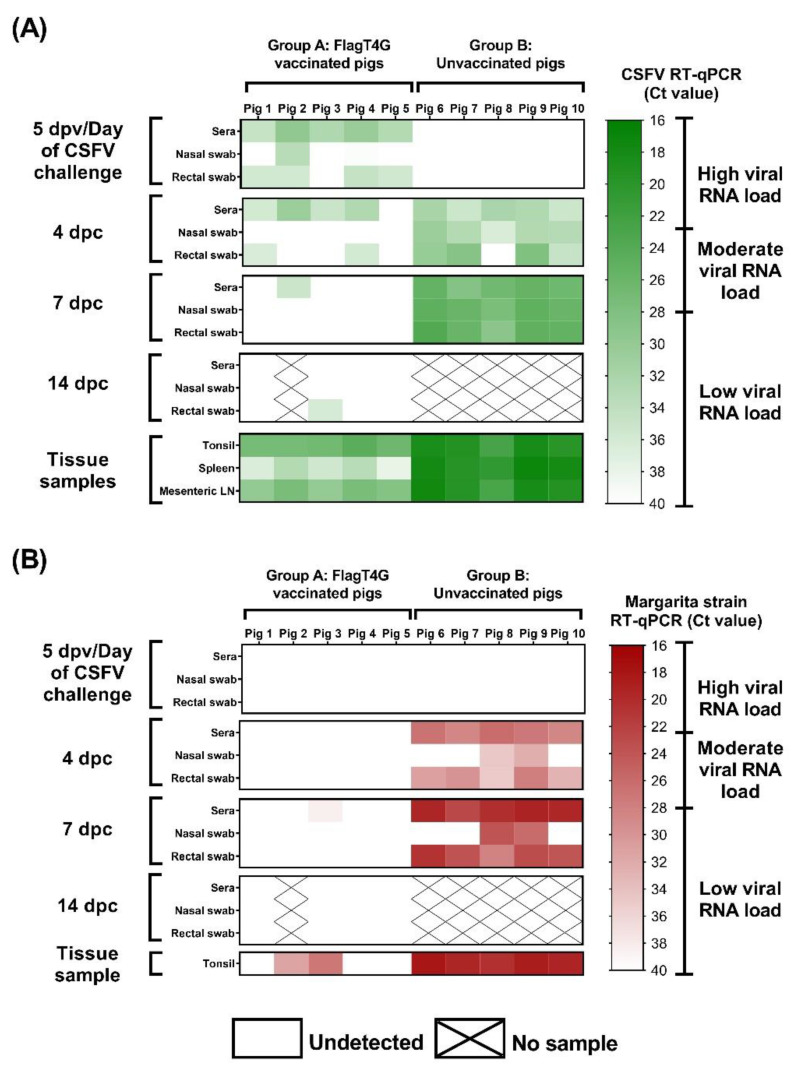
Detection of viral RNA by RT-qPCR. Samples from vaccinated and unvaccinated pigs were evaluated by the pan-CSFV (**A**) and the specific Margarita strain (**B**) RT-qPCR assays. Results are shown in a colorimetric scale representing the Ct value in green and red for the pan-CSFV and Margarita tests, respectively.

**Figure 5 viruses-14-01954-f005:**
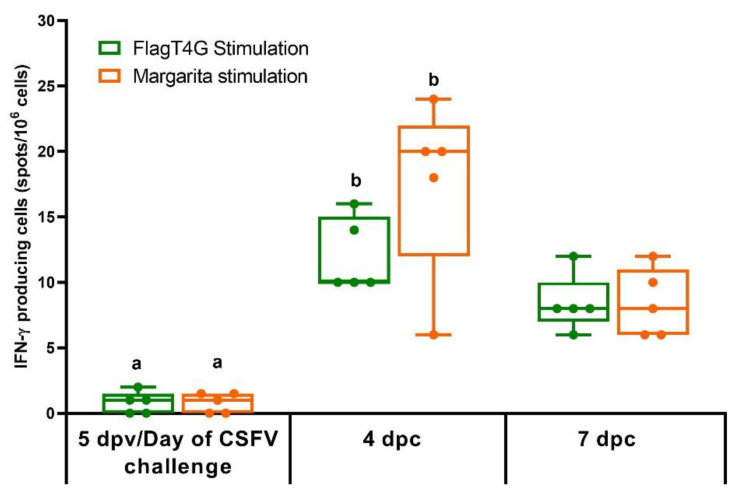
IFN-γ response in FlagT4G-vaccinated pigs. PBMCs from all the experimentally infected animals were stimulated by duplicate with the FlagT4G (green bars) and Margarita (Orange bars) viruses and evaluated by ELISPOT assay. Results are represented as the mean value of spots/10^6^ cells for each stimulation at the respective time points. Unvaccinated and challenged animals remained unresponsive to either of the viral stimuli. Differences between a given sampling and the previous one within FlagT4G-vaccinated pigs (Friedman test) are represented by different letters (*p* < 0.05).

**Figure 6 viruses-14-01954-f006:**
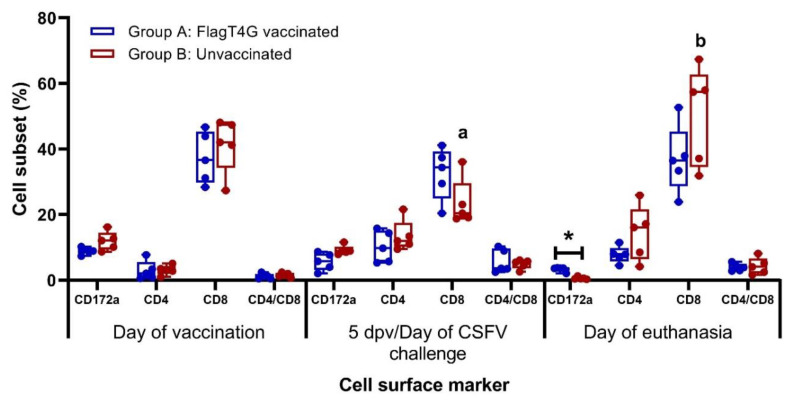
Immunological phenotypic profile after CSFV vaccination and challenge. Myelomonocytic and lymphocytic cell surface markers were evaluated in PBMC from vaccinated (blue boxes) and unvaccinated (red boxes) pigs. Mean values and cell subset percentages for each group at every sampling time are represented. Flow cytometry assays were carried out in duplicates. Asterisk indicates statistically significant differences between groups on that particular sampling (Kruskal–Wallis non-parametric test; *p* < 0.05). Differences between a given sampling and the previous one within the same group (Friedman test) are represented by the letters on top of the bar: different letters show statistical difference (*p* < 0.05).

**Table 1 viruses-14-01954-t001:** Neutralizing antibody titers against FlagT4G and Margarita strain.

Pig ID		FlagT4G				CSFV Margarita Strain			
		5 DPV/Day of CSFV Challenge	4 dpi	7 dpi	14 dpi	5 DPV/Day of CSFV Challenge	4 dpi	7 dpi	14 dpi
FlagT4G vaccinated pigs	1	(-)	(-)	1:40	1:120	(-)	(-)	(-)	1:15
	2 †	(-)	1:15	1:80	N/S	(-)	(-)	(-)	N/S
	3	(-)	1:20	1:80	1:160	(-)	(-)	1:15	1:80
	4	(-)	1:15	1:30	1:60	(-)	1:10	1:20	1:30
	5	(-)	1:10	1:30	1:240	(-)	(-)	(-)	1:10
Unvaccinated pigs	6 *	(-)	(-)	(-)	N/S	(-)	(-)	(-)	N/S
	7 *	(-)	(-)	(-)	N/S	(-)	(-)	(-)	N/S
	8 *	(-)	(-)	(-)	N/S	(-)	(-)	(-)	N/S
	9 *	(-)	(-)	(-)	N/S	(-)	(-)	(-)	N/S
	10 *	(-)	(-)	(-)	N/S	(-)	(-)	(-)	N/S

* Animals were euthanized at 7 DPC. † Animal was euthanized at 10 DPC. N/S: No sample.

## Data Availability

Not applicable.

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
