# Peer review of "The FlagT4G Vaccine Confers a Strong and Regulated Immunity and Early Virological Protection against Classical Swine Fever"

_viruses, 2022, doi:10.3390/v14091954_

Round 1

Reviewer 1 Report

The manuscript authored by Bohórquez et al. presents the immune responses induced by the CSF vaccine candidate FlagT4G and its capacity to protect pigs at a short time after vaccination. Some articles on the rapid immune protection elicited by CSF vaccines have been published.

Major comments:

1. Previous studies showed that the cellular immune response is necessary for the early protection elicited by CSF vaccines, which is distinguishable to the findings of the manuscript.

2. The data on the positive control in Figure 5 are missing.

3. The following reference should be cited: Comprehensive evaluation of the host responses to infection with differentially virulent classical swine fever virus strains in pigs. Virus Res. 2018 Aug 15;255:68-76.

Author Response

Please, see attached document 

Reviewer 2 Report

In the manuscript “The FlagT4G vaccine confers a strong and regulated innate immunity that correlates with early virological protection against Classical swine fever”, Bohórquez et al. have demonstrated that the FlagT4G live attenuated vaccine candidate is able to protect animals against the virulent CSFV Margarita as early as five days post vaccination. The results presented in this study indicate that IFNα may be a protective correlate associated with vaccination. In addition, modulation of a selection of other cytokines involved in innate immunity was studied in this work, but the results have not been sufficiently interpreted. The authors have also attempted to characterise the humoral and cellular immune responses of vaccinated and challenged animals in this study. 

While the work presented adds to the evidence base of the efficacy of the FlagT4G vaccine, there are several concerns, which if addressed, could strengthen the manuscript. Based on the current form of the manuscript, the recommendation is to review and amend the manuscript for resubmission.

Major concerns:

1. Correlation analysis of results were not performed in this study and strong and/regulated innate immunity was only demonstrated in some of the immune modulators studied, hence the title of the manuscript should be rephrased to better represent the work in this manuscript.

2. Line 159: 5 x 106 cells were seeded per well in a 96 well plate for ELIspot assays. This is an unusually high number of cells that would result in very high background and loss of linearity in results (due to multiple layers of cells in each well). Can the authors comment on this as the results obtained may not represent the true result if lower numbers of cells were used? In Figure 5, the number of spots obtained is rather low for the number of cells used, such that ‘efficient’ should not be used to describe the differences. It is strongly recommended that the authors repeat this assay with lower number of cells (between 3 x 105 and 5x 105 cells). It is unclear if statistical analysis was performed. The authors should also add the number of technical replicates assayed in the material and methods and the figure legend.

3. For the flow cytometry assays, the authors need to be careful with their labelling of the populations identified. As only CD8 markers were used, so these studies have not differentiated between NK cells that express CD8 and CD8+ T cells. In this study, the populations under the group ‘CD8 T cells’ are a mixture of CD8 expressing cells that contain both CD8+ T cells and NK cells. This is the same with the population labelled CD4 T cells, which in fact is a mixture of CD4 expressing T cells and other CD4 expressing hemopoietic cell types. The authors should make this clear in their manuscript and change the labelling used. 

4. The gating strategy used for flow cytometry analysis needs to be added into the supplementary data associated with this manuscript.

5. The results from the other cytokines assessed in this work were not discussed in the discussion. Can the authors add this to the discussion and comment on how it relates to previous studies with FlagT4G and other vaccine strains. 

Minor/specific concerns:

1. Line 263: The increase in IFNα in pig 8 is from Luminex data, but these results are not broken down in Figure 3B. Do the authors refer to the results observed in Figure 3A or is this data not shown in the manuscript? It is recommended that a breakdown of the Luminex data into responses in each animal is added to the supplementary data.

2. Figures: the statistical tests used for each set of data needs to be added to the figure legends so that it is clear for the reader which tests were used for each data set. The font sizes used for labelling and axes also needs to be increased for better readability. Consolidated data should be represented either as box-whisker plots or the data points should be shown on the graphs to represent the spread of data more accurately. A split axis or log axis will also aid interpretation of the graphs. 

3. Figure legends: The number of biological replicates and technical replicates for each data set should be added to the figure legends. 

4. Figure 3B: the resolution of the graphs needs to be increased. 

5. Line 253: What is the P value for significance in this dataset? This should be added to the figure legend.

6. It is unclear if stimulation was performed overnight for the ELIspot assays and how many technical replicates were used for the assays. For reproducibility, more detailed experimental methods need to be added into the materials and methods. 

7. Lines 300 and 414: There is a missed opportunity to further discuss the implications of have viral replication of vaccine strains in animals post vaccination. How does this compare to other vaccine strains and how does this translate in the field? Is this a desired trait or is there room for improvement?

8. Lines 360-361: so how would vaccinated and non-vaccinated animals be differentiated at such an early stage?

9. Lines 369 and 422: Can the authors clarify these statements? Were the pigs from both groups housed together? If yes, this should be clearly explained in the materials and methods.  If these statements are meant to explain something else, they should be re-written or expanded on for avoid confusion.

10. Line 380: correlation analysis was not performed in this work, so the limitations should be added into the study. Such generalised statements should be used with caution as there are other key modulators that were not assessed in this study. 

11. Figure 4: can the authors comment on the discrepancy between the results obtained with the qPCR of nasal swabs using the pan-CSFV and Margarita assays? It would also be helpful if the authors could briefly describe the two assays in the materials and methods even if the original works were referenced. 

12. Discussion: It is recommended that the authors indicate which set of data or figures are being discussed in each part of the discussion to improve readability. 

13. There is a missed opportunity to discuss the modulation of cytokines induced by the FlagT4G vaccine and other vaccine strains. It is recommended that the authors compare the results with that observed by other CSFV vaccines in use.

14. Can the authors comment on the choice of cytokines chosen for this study? What about IFNβ, IL28 and IL29?

15. It would have been interesting to have a control group that was just vaccinated and not challenged to characterise the humoral and cellular responses induced by the vaccine itself, as the results here are complicated by the challenge with virulent CSFV. Are there other studies that the authors can reference and discuss for FlagT4G? 

16. Line 95: A rather high dose of FlagT4G was used in this study. Is this commonly used and how does this compare to doses of other types of CSFV vaccines. It is recommended that the authors add this to the discussion.

17. There are some grammatical errors in the text. The authors should go through the text and correct these.

18. There are formatting errors in the references that need to be corrected.

Author Response

Please, see attached document.

Round 2

Reviewer 2 Report

In the revised manuscript The FlagT4G vaccine confers a strong and regulated immunity and early virological protection against classical swine fever, the authors have taken the comments on board, amended the manuscript and added the supplementary information as recommended. Overall, the manuscript reads much better and the discussion does help to explain how this study adds to the existing knowledge pool.

There are a few minor points that should be addressed before the manuscript can be published.

1. There is a spelling error in the caption of Table 1 and there seems to be a formatting error in the document after table 1 was added.

2. Labels on axes of Figure 3b and minor axes of Figure 4 are too small. These need to be increased for better readability.

3. Reference 8 (line 603): Is this the intended format of the reference as it has the date of access at the end?

Author Response

Please, see the attached document.
